Effects of long-term fertilisation on aggregates and dynamics of soil organic carbon in a semi-arid agro-ecosystem in China

Zhang Jiaoyang 1 3
Sun Caili 2
Liu Guobin gbliu@ms.iswc.ac.cn 1 2
Xue Sha 1 2
1 State Key Laboratory of Soil Erosion and Dryland Farming on the Loess Plateau, Institute of Soil and Water Conservation, Chinese Academy of Sciences and Ministry of Water Resources , Yangling , China
2 State Key Laboratory of Soil Erosion and Dryland Farming on the Loess Plateau, Institute of Soil and Water Conservation, Northwest A & F University , Yangling , China
3 University of Chinese Academy of Sciences , Beijing , China
Linstädter Anja
Electronic publication date: 2018 May 21
Publication date: 2018
Volume: 6
Electronic Location ID: e4758
Received 2017 Nov 12; Accepted 2018 Apr 23
Copyright: ©2018 Zhang et al.
Copyright year: 2018
Copyright holder: Zhang et al.
License: This is an open access article distributed under the terms of the Creative Commons Attribution License, which permits unrestricted use, distribution, reproduction and adaptation in any medium and for any purpose provided that it is properly attributed. For attribution, the original author(s), title, publication source (PeerJ) and either DOI or URL of the article must be cited.
License URL: https://creativecommons.org/licenses/by/4.0/

Keywords: Long-term fertilization, Path analysis, Aggregate, Volume fractal dimension, The loess Plateau

Funding: National Key Research and Development Program of China 2016YFC0501707 National Key Technology R&D Program 2015BAC01B03 West Young Scholars Project of The Chinese Academy of Sciences XAB2015A05 This work was supported by the National Key Research and Development Program of China (2016YFC0501707), the National Key Technology R&D Program (2015BAC01B03) and the West Young Scholars Project of The Chinese Academy of Sciences (XAB2015A05). The funders had no role in study design, data collection and analysis, decision to publish, or preparation of the manuscript.

==============================
Background

Long-term fertilisation has a large influence on soil physical and chemical properties in agro-ecosystems. The effects on the distribution of aggregates, however, are not fully understood. We determined the dynamic change of the distribution of aggregates and soil organic carbon (SOC) content over time in a long-term field experiment established in 1998 on the Loess Plateau of China and illustrated the relationship between them.

Methods

We determined SOC content and the distribution of aggregates in nine fertiliser treatments: manure (M); nitrogen (N); phosphorus (P); M and N; M, N, and P; M and P; N and P; bare land; and an unfertilised control. These parameters were then used for a path analysis and to analyse the fractal dimension (Dv).

Results

The organic fertiliser increased SOC content. The proportions of 0.1–0.25 mm microaggregates and 0.25–0.5 mm macroaggregates were higher and the proportion of the 0.01–0.05 mm size class of the silt + clay fraction was lower in the treatments receiving organic fertiliser (M, MN, MNP, and MP) than that in the control, indicating that the addition of organic fertiliser promoted aggregation. The distribution of aggregates characterised by their fractal dimension (Dv), however, did not differ among the treatments.

Discussion

Dv was strongly correlated with the proportion of the <0.002 mm size class of the silt + clay fraction that did not differ significantly among the treatments. The change in the distribution of aggregates was strongly correlated with SOC content, which could produce organic polymer binding agents to increase the proportion of larger particles. Long-term application of organic fertiliser is thus necessary for the improvement and maintenance of soil quality in semi-arid agricultural land when residues are removed.

Introduction

Soil organic carbon (SOC) is important for the long-term sustainability of agro-ecosystems and the environment, because it promotes aggregation, improves soil physical properties and water retention, and increases productivity and the activity of soil organisms (Paradelo, Virto & Chenu, 2015; Haynes & Naidu, 1998). Restoration of SOC content in agricultural soil represents a sink for atmospheric CO2, which has the potential to mitigate the effects of global emissions of greenhouse gases (Brar et al., 2013; Yang et al., 2003). SOC content in agriculture can be improved by the adoption of suitable management practices. Rudrappa et al. (2006) and Jaiarree et al. (2014) reported that a balanced application of mineral and organic fertilisers contributed to the increase and maintenance of SOC content in arable land.

The distribution and stability of soil aggregates are important indicators of soil physical quality, highlighting the importance of soil management on particle aggregation and disaggregation (Silva et al., 2014). Aggregate stability is generally strongly correlated with SOC content, because the cohesion of aggregates is promoted mainly by organic polymer binding agents (Haynes & Swift, 1990; Majumder, Ruehlmann & Kuzyakov, 2010) and by the physical trapping of particles by fine roots and fungal hyphae (Helfrich et al., 2015; Chenu, Le Bissonnais & Arrouays, 2000). Sequestration of SOC is mainly attributed to microaggregates (0.05–0.25 mm), because they are stabilised by persistent binding agents (>0.25 mm), and macroaggregates are stabilised by transient binding agents (Lugato et al., 2010; Yu et al., 2012).

The long-term application of organic fertiliser often increases SOC content (Yu et al., 2012; Saha, Kukal & Bawa, 2014) and the proportion of macroaggregates (Huang et al., 2010; Whalen, Hu & Liu, 2003). Reports of the effect of organic fertiliser on the distribution of microaggregates, however, have been inconsistent. Yu et al. (2012) and Tripathi et al. (2014) found that the application of organic fertiliser, compost and farmyard manure, respectively, significantly reduced the proportion of microaggregates. Interestingly, some studies have reported that organic fertiliser had no significant influence on the microaggregates relative to an unfertilised control (Chen et al., 2010; Liu et al., 2013). These different effects of organic fertiliser on the distribution of microaggregates may be attributed to the specific soil characteristics and climatic conditions (Yu et al., 2012) and to the large range of size fractions of aggregate distribution determined by the classical wet-sieving method. Few studies have subdivided the microaggregates (<0.25 mm) into smaller size fractions, which impeded a deep understanding of the changes of microaggregates in response to fertilisation.

The analysis of soil-particle distribution by laser diffraction is now commonly used (Ryzak & Bieganowski, 2011; Xiao et al., 2014) and allows the categorisation of microaggregates into smaller size classes and provides detailed volume information for each size class. Fractal theory is an effective tool for describing complex and irregular geometry (Mandelbrot, 1983). Various soils have different particle compositions with irregular shapes and self-similar structures and have fractal characteristics (Tyler & Wheatcraft, 1989). Tyler & Wheatcraft (1992) proposed the weight fractal dimension (Dm) for studying the fractal characteristics of soil structures. The calculation of Dm, however, assumes that particles of different sizes have the same density, but this assumption has been challenged by Martín & Montero (2002). This model was later developed for the volume fractal dimension (Dv) based on laser diffraction to characterise soil-particle and aggregate distribution (Chen & Zhou, 2013; Wang, Zhou & Zhao, 2005). Xiao et al. (2014) and Zhao et al. (2006) reported that Dv well described the changes in the stability of soil aggregates and in soil structure associated with vegetative succession.

Long-term experiments can provide more realistic scenarios for observing changes in soil properties and processes (Celik et al., 2010) and are thus suitable for studying the effect of fertilisation on soil quality. Some studies have determined the effect of long-term fertilisation on soil aggregates, but few studies have determined the dynamic changes of soil quality and structure for longer than 10 years. This study therefore determined SOC content and the distribution of soil aggregates of fields on the Loess Plateau in China that had been fertilised for 15 years to demonstrate the long-term trend of SOC content and fractal dimension. We hypothesised that the long-term application of organic or mineral fertilisers could significantly influence SOC content and the distribution of soil aggregates over time. The specific objectives were thus: (1) to observe the effect of fertilisation on SOC content, aggregate distribution, and Dv and illustrate the relationship between them, and (2) to describe the dynamic changes in SOC content and Dv over 15 years.

Materials & Methods

Experimental site

This study was part of an on-going long-term field fertilisation experiment established in 1998 at the Ansai National Field Scientific Observation and Research Station for Farmland Ecosystems, Shaanxi province, China (36°51′30″N, 109°19′23″E). The station is at an altitude of 1,068–1,309 m a.s.l. and has a temperate semi-arid climate with a mean annual temperature of 8.8 °C and a mean annual rainfall of 500 mm. The soil is a Huangmian soil, which is classified as a Calcic Cambisol (FAO/UNESCO/ISRIC, 1988), originating from wind-deposited loessial parental material and characterised by yellow particles, an absence of bedding, a silty texture, looseness, macroporosity, and wetness-induced collapsibility (Zhu et al., 2010). The basic soil characteristics to a depth of 20 cm were: organic-matter content of 15.54 g kg−1, total nitrogen (N) content of 0.57 g kg−1, total phosphorus (P) content of 0.63 g kg−1, available N content of 28.99 mg kg−1, available P content of 2.49 mg kg−1, available potassium (K) content of 84.86 mg kg−1, pH 8.6, and bulk density of 1.5 g cm−3.

Sampling and processing

The long-term experiment had a triplicate randomised complete block design with an area of 14 m2 for each plot. Each block contained nine treatments: manure (M), N, P, M and N (MN), M, N, and P (NMP), M and P (MP), N and P (NP), unfertilised bare land (BL), and an unfertilised control (CK) (Fig. 1). BL had not been sown or fertilised, and CK was sown but not fertilised. N was added as urea, P was added as superphosphate, and the farmyard manure consisted of the faeces and urine from domestic sheep. The contents of organic matter, N, and P in the faeces were 25.7, 0.75, and 0.54%, respectively. The contents of N and P in the urine were 1.4 and 0.45%, respectively. The amounts of the fertilisers applied in the treatments are presented in Table 1. P, M, and 20% of the N were applied together as seed fertilisers, and the remaining 80% of the N was top-dressed between the large-bell and tasselling stages. The experiment had a three-year rotation, with a sequence of Glycine max-Zea mays-Z. mays, beginning with G. max in the autumn of 1998. The last crops of Z. mays (cv. QiangSheng 101) were seeded on 29 April 2012 at a rate of 52.5 kg ha−1, and the plant density was about 51,000 plants ha−1. The crops were manually harvested, and the aboveground residues were removed in October.

Figure 1 Scheme showing the experimental area with various fertiliser treatments.

Note: Mineral fertilizers were nitrogen (N) and phosphorus (P), organic fertilizer was farmyard manure (M). Treatments were mineral fertilizers, organic manure, different combinations of mineral fertilizers and organic manure, unfertilized bare land (BL) and an unfertilized control (CK).

Table 1 The amounts of fertilisers applied in long-term experiment (kg ha−1).

Treatments	Organic manure	N fertiliser	P fertiliser	
BL	0	0	0	
CK	0	0	0	
M	7,500	0	0	
MN	7,500	211.95	0	
MNP	7,500	211.95	166.65	
MP	7,500	0	166.65	
N	0	211.95	0	
NP	0	211.95	166.65	
P	0	0	166.65	
Notes.

N was added as urea, P as superphosphate, and the farmyard manure contained the feces and urine from domestic sheep. BL represents bare land (no plants nor fertilisers) and CK represents unfertilised control. Contents of organic matter, N and P in feces were 25.7, 0.75, and 0.54% respectively. Contents of N and P in urine were 1.4 and 0.45% respectively.

Sampling

The soil was sampled annually in October from 1998 to 2012 to a depth of 20 cm. Three replicate soil samples were randomly excavated in each plot using a soil drill (diameter, 4 cm) and then mixed to produce a composite sample. The samples for 1998–2012 were used for the analysis of changes over time. The samples for 1998–2011 (126 samples) had been stored in the station’s soil library and were collected from one of the three replicate treatment plots. The samples for 2012 (27 samples) were used to examine the effects of fertilisation on aggregate distribution, Dv, and SOC content. Visible plant residues were removed, and the samples were manually broken into fragments <10 mm and air-dried at room temperature. Each sample was passed through a 1-mm sieve for determining the distribution of the aggregates, and a subsample was then ground to pass through a 0.25-mm sieve for the determination of total SOC content.

Determination of aggregate distribution

The soil samples were soaked in distilled water for 24 h and mechanically dispersed by ultrasonication for 5 min (Xiao et al., 2014). The samples were analysed with a Longbench Mastersizer 2000 (Malvern Instruments, Malvern, England).

Determination of SOC content

SOC content was determined by Walkley and Black dichromate oxidation (Nelson & Sommers, 1982).

Dv of aggregates

Dv was calculated as:

(1) Vr<RiVT=RiRmax3−Dv

where r is the particle diameter, Ri is the diameter of size class i, V (r<Ri) is the total volume of particles with diameters <Ri, VT is the total volume of particles, Rmax is the maximal particle diameter, and Dv is the volume fractal dimension. Logarithms were derived for both sides of the equation, and Dv was obtained from the slopes of the double-logarithmic curves that fit the data.

Path analysis

Path analysis is a supplement and extension of regression analysis that partitions simple correlation coefficients into direct and indirect effects through definite path coefficients among the variables and distinguishes between correlation and causation (Bai et al., 2014; Wright, 1934; Zhang et al., 2005).

The aggregates were categorised into nine size classes: <0.002, 0.002–0.005, 0.005–0.01, 0.01–0.05, 0.05–0.1, 0.1–0.2, 0.2–0.25, 0.25–0.5, and 0.5–1 mm. These nine size classes were categorised into three fractions: macroaggregates (0.25–1 mm), microaggregates (0.05–0.25 mm), and a silt + clay fraction (<0.05 mm). Backward-stepwise regression was used to identify the size class that explained most of the variation in Dv, with criteria for the stepwise regression of probability-of-F-to-enter ≤0.05 and probability-of-F-to remove ≥0.10. The size classes that did not significantly contribute to Dv at P = 0.10 were therefore not included in the regression model.

Path diagrams were used to evaluate the relationships between eight selected size classes and Dv (Fig. 2). The direct effects of the aggregates on Dv are represented by single-headed arrows, and the coefficients of the correlations between the size classes are represented by double-headed arrows. The direct effects are termed path coefficients and are standardised partial regression coefficients in the multiple linear regression of aggregates on Dv (Basta, Pantone & Tabatabai, 1993). The indirect effects are determined from the product of the simple correlation coefficient between aggregate size classes and the path coefficient. The results of the path analysis were determined from the equations (Williams, Demment & Jones, 1990):

(2) r19=P19+r12P29+r13P39+r14P49+r15P59+r16P69+r17P79+r18P89

(3) r29=r12P19+P29+r23P39+r24P49+r25P59+r26P69+r27P79+r28P89

(4) r39=r13P19+r23P29+P39+r34P49+r35P59+r36P69+r37P79+r38P89

(5) r49=r14P19+r24P29+r34P39+P49+r45P59+r46P69+r47P79+r48P89

(6) r59=r15P19+r25P29+r35P39+r45P59+P59+r56P69+r57P79+r58P89

(7) r69=r16P19+r26P29+r36P39+r46P49+r56P59+P69+r67P79+r68P89

(8) r79=r17P19+r27P29+r37P39+r47P49+r57P59+r67P69+P79+r78P89

(9) r89=r18P89+r28P29+r38P39+r48P49+r58P59+r68P69+r78P79+P89

where rij is the simple correlation coefficient between aggregate size class and Dv, Pij is the path coefficient between aggregate size class and Dv (direct effects), and rijPij is the indirect effect of aggregate size class on Dv. Subscript designations 1–9 represent the <0.002, 0.002–0.005, 0.005–0.01, 0.05–0.1, 0.1–0.2, 0.2–0.25, 0.25–0.5, 0.5–1 mm aggregates and Dv, respectively.

Figure 2 Path analysis diagram for the relationships between fractal dimension and micro-aggregate fractions.

Note: Direct path coefficients (Pij) of micro-aggregate fractions are presented by single-headed arrows while simple correlation coefficients (rij) between variables are represented by double-headed arrows. Subscript designations of 1–9 are <0.002, 0.002–0.005, 0.005–0.01, 0.05–0.1, 0.1–0.2, 0.2–0.25, 0.25–0.5, 0.5–1 mm micro-aggregates, and fractal dimension (Dv), respectively.

The residual, U, is an unmeasured variable in the path model that represents the unexplained part of an observed variable and is calculated as (Ige, Akinremi & Flaten, 2007):

(10) U=1−R2

where R2 is the coefficient of determination of the multiple regression model between aggregate size classes and Dv.

The coefficient of determination for each factor denotes the degree of relative determination between cause and effect, which can be determined by (Bai et al., 2014; Chen et al., 2014):

(11) Dyxixj=Pyi2,i=j;Dyxixj=2Pyi×Pyj×rij,i,j=1,2,…,n,i<j

where Dyxixj is the coefficient of determination, and y is Dv.

Statistical analyses

One-way analyses of variation tested the differences between the various fertilisation treatments. Pearson correlation coefficients were calculated to analyse the relationships among aggregates, Dv, and SOC content. Duncan tests separated the means of these variables at P < 0.05. All statistical analyses were conducted using the R statistical package (version 3.1.0) (R Core Team, 2014).

Results

Effects of fertilisation treatments on SOC content, aggregate distribution, and Dv

SOC content was significantly higher in the treatments receiving organic fertiliser (M, MN, MP, and MNP) than in the treatments receiving mineral fertiliser (N, NP, and P) or in BL or CK. SOC content did not differ significantly among BL, CK, N, NP, and P. The 0.01–0.05 size class of the silt + clay fraction and 0.05–0.1 mm microaggregates represented >50% of the total soil aggregates (Table 2). The proportion of the 0.01–0.05 mm size class of the silt + clay fraction was the lowest in M, and the proportion of 0.1–0.25 mm microaggregates was significantly higher in M than CK. The proportion of 0.25–0.5 mm macroaggregates was significantly higher in the treatments receiving organic fertiliser (M, MN, MNP, and MP) than CK. In contrast, the distribution of aggregates in the treatments receiving mineral fertilisers (N, NP, and P) and BL did not differ significantly from that in CK. Dv did not differ significantly among the nine treatments.

Table 2 Effects of fertilizer treatments on aggregates, fractal dimension (Dv) and soil organic carbon content (SOC).

Treatments	<0.002 mm	0.002–0.005 mm	0.005–0.01 mm	0.01–0.02 mm	0.02–0.05 mm	0.05–0.1 mm	0.1–0.2 mm	0.2–0.25 mm	0.25–0.5 mm	0.5–1 mm	Dv	SOC (%)	
BL	4.13 ± 0.29	6.11 ± 0.55	7.1 ± 0.56	10.17 ± 0.69ab	33.91 ± 0.54ab	27.46 ± 1.11ab	7.46 ± 0.65bcd	0.37 ± 0.07bc	2.14 ± 0.37ab	1.14 ± 0.37	2.736 ± 0.00	0.58 ± 0.05d	
CK	4.42 ± 0.01	6.44 ± 0.14	7.31 ± 0.21	10.52 ± 0.25ab	34.90 ± 0.84a	27.04 ± 0.77ab	7.08 ± 0.13cd	0.38 ± 0.07bc	1.36 ± 0.52b	0.56 ± 0.28	2.736 ± 0.00	0.65 ± 0.03cd	
M	4.00 ± 0.3	5.97 ± 0.59	6.93 ± 0.47	9.73 ± 0.37b	32.74 ± 0.94c	27.87 ± 1.32ab	8.79 ± 0.25a	0.81 ± 0.31a	2.31 ± 0.21a	0.87 ± 0.16	2.735 ± 0.01	0.97 ± 0.08a	
MN	4.09 ± 0.39	6.05 ± 0.71	7.10 ± 0.61	10.27 ± 0.6ab	33.92 ± 0.24ab	26.93 ± 0.91ab	7.76 ± 0.39bc	0.58 ± 0.05abc	2.23 ± 0.57a	1.05 ± 0.47	2.735 ± 0.01	0.90 ± 0.01ab	
MNP	4.00 ± 0.15	5.94 ± 0.33	6.95 ± 0.24	9.96 ± 0.15b	33.74 ± 0.59bc	27.3 ± 0.36ab	7.99 ± 0.15ab	0.67 ± 0.2ab	2.39 ± 0.46a	1.05 ± 0.12	2.734 ± 0.01	0.87 ± 0.03b	
MP	3.99 ± 0.16	5.93 ± 0.31	6.95 ± 0.43	10.00 ± 0.63b	33.62 ± 0.31bc	27.19 ± 0.85ab	8.03 ± 0.29ab	0.70 ± 0.24ab	2.41 ± 0.45a	1.18 ± 0.59	2.734 ± 0.00	0.93 ± 0.06ab	
N	4.08 ± 0.43	5.97 ± 0.79	7.03 ± 0.66	10.26 ± 0.55ab	34.58 ± 0.23ab	27.61 ± 0.89ab	7.33 ± 0.48bcd	0.30 ± 0.16c	1.90 ± 0.44ab	0.94 ± 0.27	2.732 ± 0.01	0.63 ± 0.04cd	
NP	4.30 ± 0.18	6.43 ± 0.36	7.7 ± 0.63	11.23 ± 1.05a	34.59 ± 0.56ab	26.22 ± 1.11b	6.82 ± 0.81d	0.30 ± 0.19c	1.70 ± 0.37ab	0.71 ± 0.27	2.735 ± 0.00	0.67 ± 0.05c	
P	4.03 ± 0.25	5.9 ± 0.47	6.81 ± 0.41	9.71 ± 0.33b	33.99 ± 0.29ab	28.45 ± 0.47a	8.13 ± 0.41ab	0.43 ± 0.13bc	1.82 ± 0.56ab	0.72 ± 0.38	2.732 ± 0.01	0.67 ± 0.02c	
F value	0.906	0.509	0.856	1.963	3.966*	1.409	5.456*	3.372*	1.907	1.123	0.279	32.983*	
CV (%)	3.42	3.25	3.54	4.30	1.78	2.16	7.39	35.43	16.62	22.08	0.05	18.64	
Notes.

* significant at P < 0.01.

Correlations among aggregates, Dv, and SOC content

The proportions of all size classes of the silt + clay fraction were positively correlated with each other, except between the 0.01–0.05 mm and <0.002 mm and 0.005–0.01 mm size classes, and were negatively correlated with the proportions of micro- and macroaggregates (0.05–1 mm), except between the 0.2–0.25 and 0.002–0.005 and 0.01–0.05 mm size classes (Table 3). The proportion of 0.05–0.1 mm microaggregates was positively correlated with the proportions of 0.1–0.2 mm microaggregates and macroaggregates (0.25–1 mm), the proportion of 0.1–0.2 mm microaggregates was positively correlated with the proportion of macroaggregates, and the proportion of 0.25–0.5 mm macroaggregates was positively correlated with the proportion of 0.5–1 mm macroaggregates. Dv was correlated positively with the proportion of the <0.01 mm size class of the silt + clay fraction and negatively with the proportions of the 0.01–0.05 mm size class of the silt + clay fraction and 0.2–0.25 mm microaggregates. SOC content was correlated negatively with the proportion of the <0.05 mm size class of the silt + clay fraction and positively with the proportions of micro- and macroaggregates (0.05–1 mm).

Table 3 Correlations relationship of aggregates, fractal dimension (Dv) and soil organic carbon content (SOC).

Fractions	Silt and clay fraction	Microaggregates	Macroaggregates	Dv	SOC (%)	
Particle size	<0.002 mm	0.002–0.005 mm	0.005–0.01 mm	0.01–0.02 mm	0.02–0.05 mm	0.05–0.1 mm	0.1–0.2 mm	0.2–0.25 mm	0.25–0.5 mm	0.5–1 mm			
<0.002 mm	1												
0.002–0.005 mm	0.844**	1											
0.005–0.01 mm	0.826**	0.834**	1										
0.01–0.02 mm	0.529**	0.597**	0.701**	1									
0.02–0.05 mm	0.075	0.235**	0.020	0.624**	1								
0.05–0.1 mm	−0.421**	−0.657**	−0.478**	−0.803**	−0.732**	1							
0.1–0.2 mm	−0.353**	−0.520**	−0.313**	−0.748**	−0.865**	0.881**	1						
0.2–0.25 mm	−0.266**	−0.080	−0.220**	−0.080	0.138	−0.263**	0.075	1					
0.25–0.5 mm	−0.439**	−0.358**	−0.416**	−0.503**	−0.446**	0.185*	0.237**	0.129	1				
0.5–1 mm	−0.474**	−0.454**	−0.423**	−0.589**	−0.584**	0.396**	0.413**	−0.064	0.861**	1			
Dv	0.756**	0.646**	0.623**	0.051	−0.467**	−0.106	0.055	−0.203*	0.165*	0.142	1		
SOC (%)	−0.244**	−0.177*	−0.155	−0.358**	−0.434**	0.207*	0.490**	0.500**	0.340**	0.310**	0.098	1	
Notes.

* significant at P < 0.05.

** significant at P < 0.01.

Path analysis for aggregates and Dv

A U (uncorrected residue) of 0.054 and an R2 of 0.997 indicated a small unexplained part of the observed variable in the path model, and the path analysis explained 99.7% of the variability associated with Dv (Table 4). The path coefficients (underlined in Table 4) indicated that all selected aggregate classes had significant direct effects on Dv (P < 0.01). The magnitude of the path coefficients indicated that silt + clay fraction (<0.002 mm) (P19 = 0.887) was the most important causal factor in predicting Dv, followed by the 0.002–0.005 mm size class of the silt + clay fraction (P29 = 0.368). The direct effect of the <0.002 mm size class of the silt + clay fraction (0.724) on Dv was larger than its total indirect effects (−0.132), and the direct and total indirect effects of the 0.002–0.005 mm size class of the silt + clay fraction on Dv were comparable (0.368 and 0.277, respectively). The total indirect effects of the 0.005–0.01 mm size class of the silt + clay fraction (0.586) on Dv, however, were larger than the direct effects (0.055). The indirect effects of the 0.002–0.01 mm size classes of the silt + clay fraction on Dv due to the <0.002 mm size class were large (r12P19 = 0.749 and r13P19 = 0.732).

Table 4 Direct effects (diagonal, underlined) and indirect effects (off-diagonal) of aggregates on fractal dimension (Dv).

Variable	r	<0.002 mm	0.002–0.005 mm	0.005–0.01 mm	0.05–0.1 mm	0.1–0.2 mm	0.2–0.25 mm	0.25–0.5 mm	0.5–1 mm	Total	U	R2	
<0.002 mm	0.756**	0.887**	0.311	0.045	−0.123	−0.039	−0.030	−0.167	−0.130	−0.132	0.054	0.997	
0.002–0.005 mm	0.646**	0.749	0.368**	0.046	−0.191	−0.057	−0.009	−0.136	−0.124	0.277			
0.005–0.01 mm	0.623**	0.732	0.307	0.055**	−0.139	−0.034	−0.025	−0.158	−0.116	0.586			
0.05–0.1 mm	−0.106	−0.374	−0.242	−0.026	0.291**	0.096	−0.029	0.070	0.108	−0.396			
0.1–0.2 mm	0.055	−0.313	−0.191	−0.017	0.256	0.109**	0.008	0.090	0.113	−0.054			
0.2–0.25 mm	−0.203*	−0.236	−0.030	−0.012	−0.077	0.008	0.112**	0.049	−0.017	−0.314			
0.25–0.5 mm	0.165*	−0.389	−0.132	−0.023	0.054	0.026	0.014	0.380**	0.236	−0.214			
0.5–1 mm	0.142	−0.421	−0.167	−0.023	0.115	0.045	−0.007	0.327	0.274**	−0.131			
Notes.

* significant at P < 0.05.

** significant at P < 0.01.

r correlation coefficients between micro-aggregate fractions and fractal dimension

Total total indirect path coefficient

U uncorrelated residue

The <0.002 mm size class of the silt + clay fraction had the highest coefficient of determination (Dyx1x1 = 0.787), followed by correlations between the <0.002 mm size class of the silt + clay fraction and 0.25–0.5 mm macroaggregates (Dyx1x7 = 0.674), the <0.002 and 0.002–0.005 mm size classes of the silt + clay fraction (Dyx1x2 = 0.653), and the <0.002 and 0.01–0.05 mm size classes of the silt + clay fraction (Dyx1x8 = 0.486). The other coefficients of determination were low (Table 5).

Table 5 The determination coefficients of each factor.

Fractions	Silt and clay fraction	Microaggregates	Macroaggregates	
Particle size	<0.002 mm	0.002–0.005 mm	0.005–0.01 mm	0.05–0.1 mm	0.1–0.2 mm	0.2–0.25 mm	0.25–0.5 mm	0.5–1 mm	
<0.002 mm	0.787	0.653	0.098	0.516	0.193	0.199	0.674	0.486	
0.002–0.005 mm		0.135	0.040	0.214	0.080	0.082	0.280	0.202	
0.005–0.01 mm			0.003	0.032	0.012	0.012	0.042	0.030	
0.05–0.1 mm				0.085	0.063	0.065	0.221	0.159	
0.1–0.2 mm					0.012	0.024	0.083	0.060	
0.2–0.25 mm						0.013	0.085	0.061	
0.25–0.5 mm							0.144	0.208	
0.5–1 mm								0.075	

Changes of SOC content and Dv over time

The path graph clearly identified the changes of SOC content and Dv from 1998 to 2012 (Fig. 3). SOC content tended to increase over time in the treatments receiving organic fertiliser (M, MN, MNP, and MP) but not in the treatments receiving mineral fertilisers or in BL or CK. Dv did not change over time except in BL, where it tended to increase.

Figure 3 Changes of fractal dimension and SOC content in different fertilizer treatments from 1998 to 2012.

Discussion

Effect of fertilisation on SOC content, aggregate distribution and Dv

SOC content did not differ significantly between CK and BL and the treatments receiving mineral fertiliser. The 15-year application of organic fertiliser significantly increased SOC content relative to the treatments receiving mineral fertiliser. The plots in this study were conventionally tilled, and the soybean and maize straw was removed when the crops were harvested, indicating that the increase in SOC content was mostly due to the application of organic fertiliser, which has abundant humic material that improves SOC content, physical properties, and other aspects of the soil such as N and P contents (Haynes & Naidu, 1998). The application of organic fertiliser can also stimulate the development of roots in regions with poor soil, such as the Loess Plateau (Banger et al., 2009); up to 40% of newly photosynthesised carbon is released into soil by roots, thereby increasing the pool of active organic carbon (Kuzyakov & Cheng, 2001).

Long-term fertilisation, especially with organic fertiliser, can have a large influence on the distribution of soil aggregates in agro-ecosystems (Miao, Qiao & Zhou, 2009; Plaza-Bonilla, Alvaro-Fuentes & Cantero-Martinez, 2013; Tripathi et al., 2014). Application of M alone or in combination with mineral fertiliser (MN, MNP, and MP) increased the proportions of 0.1–0.25 mm microaggregates (even though the increases were not significant among MN, MNP, and MP) and 0.25–0.5 mm macroaggregates and correspondingly decreased the proportion of the 0.01–0.05 mm size class of the silt + clay fraction. In contrast, the distribution of aggregates relative to CK was not significantly affected by the treatments with only mineral fertiliser. The application of organic fertiliser was thus quite conducive to the aggregation of soil particles. Our results were consistent with previous results by Miao, Qiao & Zhou (2009) who reported that the continual addition of organic fertiliser usually increased SOC content and microbial activity and had a positive effect on aggregation on the Songnen Plain in northwestern China. Tripathi et al. (2014), however, reported a decrease in the proportion of microaggregates under 41 years of fertilisation in a tropical agro-ecosystem in China. A study conducted in Nanchang reported that organic fertiliser did not significantly affect microaggregates (Liu et al., 2013). These different results of the distribution of microaggregates may be attributed to the specific soil characteristics and climatic conditions. The application of mineral fertilisers, however, can have little impact on SOC content unless used in conjunction with no tillage and residue management (Lal, 2004; Rudrappa et al., 2006; Yang et al., 2003). One study even suggested that the long-term application of mineral fertilisers would likely degrade small macroaggregates into microaggregates, or even into the silt and clay fraction, and can lead to disaggregation (Chen et al., 2010).

Dv could not distinguish among the various treatments, even though it can well describe the changes in the stability of aggregates and soil structure associated with vegetative succession (Xiao et al., 2014; Zhao et al., 2006), because Dv was positively correlated with the proportion of the <0.002 mm size class of the silt + clay fraction (Table 3) that did not differ significantly among the treatments (Table 2). The proportions of the 0.002–0.01 mm size classes of the silt + clay fraction were also strongly correlated with Dv, but these correlations were mostly partitioned to the indirect effect of the 0.002–0.01 mm size classes on Dv due to the <0.002 mm size class. The path analysis indicated that the total indirect effect of the 0.002–0.01 mm size classes of the silt + clay fraction on Dv was mainly due to the <0.002 mm size class, with coefficients of 0.732 and 0.749, respectively. The higher direct effect of the <0.002 mm size class of the silt + clay fraction on Dv (0.887) and the total indirect effect of the 0.002–0.01 mm size classes on Dv due to the <0.002 mm size class matched the highest positive correlation coefficient between the <0.002 mm size class and Dv, corresponding to the highest coefficient of determination of the <0.002 mm size class (Dyx1x1 = 0.787), the large coefficients of determination of the correlations between the <0.002 mm size class of the silt + clay fraction and 0.25–0.5 mm macroaggregates (Dyx1x7 = 0.674), between the <0.002 and 0.002–0.005 mm size classes of the silt + clay fraction (Dyx1x2 = 0.653), and between the <0.002 and 0.01–0.05 mm size classes of the silt + clay fraction (Dyx1x8 = 0.486). Our result was in agreement with previous studies by Tang et al. (2013) and Zhao et al. (2006). Xiao et al. (2014) also demonstrated that Dv was positively correlated with the proportion of the <0.002 mm size class of the silt + clay fraction.

Correlations between aggregates, Dv, and SOC content

The proportion of the silt + clay fraction (<0.05 mm) was generally negatively correlated with the proportions of micro- and macroaggregates (0.05–1 mm), but the proportions of some size classes in the 0.05–1 mm fractions (micro- and macroaggregates) were positively correlated, indicating that larger aggregates formed at the expense of the silt and clay fraction and vice versa and that aggregates >0.05 mm may stimulate each other to reaggregate. This phenomenon was demonstrated by Su et al. (2006), who found that long-term application of organic fertiliser significantly increased the proportions of both >2 and 0.25–2 mm aggregates. Chen et al. (2010) demonstrated that the application of mineral fertiliser increased the proportion of <0.25 mm microaggregates and decreased the proportion of 0.25–2 mm macroaggregates.

SOC content in our study was correlated negatively with the proportion of the silt + clay fraction (<0.05 mm) and positively with the proportions of micro- and macroaggregates (0.05–1 mm), indicating that the increase in SOC content facilitated aggregation and that micro- and macroaggregates played an important role in stabilising the SOC content (Liu et al., 2015; Tisdall & Oades, 1982). Mucilaginous substances released from organic fertilisers bind soil particles into microaggregates and then into macroaggregates, which would increase the proportions of the 0.1–0.25 mm and 0.25–0.5 size classes (Tisdall & Oades, 1982; Haynes & Naidu, 1998; Tripathi et al., 2014). The long-term application of organic fertiliser also often increases crop yield and above- and belowground biomass (Manna et al., 2007). Compounds produced by roots and fungal hyphae, such as polysaccharides and other byproducts generated by the decomposition of organic material in soil, can bind microaggregates together into macroaggregates (Liao et al., 2006).

Changes of Dv and SOC content over time

The long-term application of organic fertiliser increased SOC content from 1998 to 2012, in agreement with the findings by Xu et al. (2016) and Banger et al. (2009), who attributed this increase to the long-term addition of organic material and the continuous return of large amounts of biomass in the form of roots and stubble. Su et al. (2006) found that the combination of organic and mineral fertilisers could substantially increase N, P, and K contents. Crop residues were removed in our study, so the long-term application of organic fertiliser may be the most effective option for improving and maintaining nutrient levels and soil quality.

Conclusions

Treatments receiving organic fertiliser increased the proportions of 0.1–0.25 mm microaggregates and 0.25–0.5 mm macroaggregates and correspondingly decreased the proportion of the 0.01–0.05 mm size class of the silt + clay fraction relative to CK, indicating that the application of organic fertiliser was favourable to the formation of larger aggregates. Dv could not distinguish among the various treatments, because Dv was mainly determined by the proportion of the <0.002 mm size class of the silt + clay fraction that did not differ significantly among the treatments. SOC content was significantly higher in the treatments receiving organic fertiliser (M, MN, MP, and MNP) compared to those receiving mineral fertiliser, BL, and CK and tended to increase over time, which promoted the formation of larger aggregates and the sequestration of SOC. The application of organic fertiliser can thus contribute greatly to the improvement and sustainability of soil quality in semi-arid agricultural land when residues are removed.

Supplemental Information

Data S1 Raw data of the analyses

Click here for additional data file.

We thank the anonymous reviewers and the journal editors for providing constructive comments and suggestions on the manuscript.

Additional Information and Declarations

Competing Interests

Author Contributions

Data Availability

The authors declare there are no competing interests.

Jiaoyang Zhang analyzed the data, contributed reagents/materials/analysis tools, prepared figures and/or tables.

Caili Sun performed the experiments, contributed reagents/materials/analysis tools.

Guobin Liu conceived and designed the experiments.

Sha Xue conceived and designed the experiments, authored or reviewed drafts of the paper, approved the final draft.

The following information was supplied regarding data availability:

The raw data are included in Data S1.

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
