# Peer review of "Effects of long-term fertilisation on aggregates and dynamics of soil organic carbon in a semi-arid agro-ecosystem in China"

_PeerJ, doi:10.7717/peerj.4758_

## Round 0.1 · original submission · Major Revisions

Your study takes advantage of a long-term fertilization experiment in China and of annual soil sampling to study the effects of fertilization on (i) the distribution of soil aggregates and (ii) on carbon sequestration over 15 years (1998-2012). You hypothesised that a long-term application of organic or mineral fertilisers could improve SOC content and the distribution of soil aggregates. Results show that only treatments receiving organic fertilisers increased SOC content, while no change was observable for treatments receiving mineral fertilisers. Organic fertilisers also altered aggregate structure.

Your manuscript is technically sound and well-written. However it received mixed reviews. While two reviewers found your paper convincing and only asked for minor revisions, Reviewer 2 questions the novelty of your study. I agree with this reviewer that the research gap and, thus, the novel aspect of your study should be much more clearly developed. Results also need to be discussed with respect to existing scientific knowledge.

In my point of view, the most interesting aspect of your study is that for the question of how soil organic carbon (SOC) is sequestered, it might also be important to consider indirect effects of fertilisation via its effects on microaggregate structure. Accordingly, the main research gap addressed in your study is if fertiliser application might (also) have effects on the distribution of microaggregates. While you mention this gap in your introduction, you do not pick this issue up in your discussion in a convincing way. Moreover, more specific hypotheses are required for your semi-arid site, sustained by a literature review of what other fertilization studies found for dryland sites and for more humid sites. Here I fully agree with Reviewer 3 that you need to sum up the related studies, and compare the differences in the results of arid/semi-arid areas and more humid regions.

Finally, the abstract is also not really instructive and appealing – e.g. the research gap should be stated here too. The interesting long-term dataset (highlighted e.g. by Reviewer 2) is also worth to be mentioned more explicitly (i.e. mention the duration of your study and the fact that annual samples were analysed).

Specific comments:

Figure captions: Please make captions self-explanatory. For example the caption of Fig. 3 (“Changes of fractal dimension and SOC content over time”) gives not enough information to interpret the graph. I would suggest a colour code to visualize the temporal changes over the 15-year period!

Table captions: Like figure captions, please make them self-explanatory. For example for Table 1, treatment labels should be explained in the caption (incidentally; the label “CK” for the unfertilised control is not really convincing as there is no “K” in the long name).

Table 5 (determination coefficients of each factor): Please also give the aggregate names in the table itself, as it is difficult to match the particle size classes to the respective aggregate names.

Page 11: “The path coefficients (underlined in Table 4) indicated that all selected aggregate classes had significant direct effects and on Dv (P<0.01)”: Remove the word “and”

Reviewer 1 ·

Basic reporting

Author’s understanding about organic fertilizer is not correct.
When there are N and P from the application of urea and superphosphate, respectively, I do not think that author can write these under organic fertilizer. It is a mix of organic and mineral fertilizers. Check: http://www.sciencedirect.com/topics/agricultural-and-biological-sciences/organic-fertilizer.

Experimental design

Well-presented experimental design section.
Lines #19-20: Missing- why author is testing nine fertilizer treatments? A complete sentence is required. PS: this is a part of your abstract.
Author is encouraged to include an information about weather for the whole study period and show its relation to aggregate distribution and SOC content under Discussion section.
Line #108: Have you tested the major nutrient constituents of farm-yard manure derived from sheep? If yes, report it.

Validity of the findings

Few suggestion-
Besides table 4 and table 5- can you add a figure which can represent aggregate size distribution based on your nine different treatments?
Fig.3: Hard to read- choice of color for ‘year’ is not ideal. This needs to revise.

Additional comments

Lines #57-58: better to mention what was/ were the organic fertilizer in those study
Lines #104-105: 14 sq. m is total area or size of an individual plot? It’s clear on Fig. 1 but you have to make clear here also.
Line #118: 1998-2012 is 14 years, not a 15 (you are mentioning 15 years in Line #83)
Line #152: [suggestion] is there any simplified form to show regression?
Line #194: correction ‘was the lowest’
Line #219: ‘U’, let’s write uncorrelated residue too

Reviewer 2 ·

Basic reporting

The topic is not interesting because the role of fertilisation, special organic material, in improving soil aggregation and SOC is accepted.
The article is written using poor English . There are many unscientific expression.

Experimental design

The experimental design is very common. For example, the M treatment is 7500 kg/ha organic manure, N treatment with 211.95 kg N/ ha. There is different amount of N between M and N treatments. So the improve in SOC or aggregate shoud not only attribute to organic manure.

Validity of the findings

The conclusion is widely accepted result.

Additional comments

The data of long-term experiment is valuable. If possible, you'd better sum up these data and suggest that long-term fertilization is more meaningful to agriculture or the agricultural economy.

·

Basic reporting

It is significant to study the effect of Long-term fertilization on aggregates and dynamics of soil organic carbon in semi-arid agro-ecosystem. The structure of the article is reasonable and the chart is clear.
But the "Introduction" part should sum up the related studies in the arid and semi-arid area, and compare the differences in the results of arid/semi-arid areas and other regions.

Experimental design

The experimental design is reasonable and the result is reliable.

Validity of the findings

A lot of research in this field, the article should highlight its characteristics in semi-arid region, other results can be classified as a tropical region, temperate humid region and arid and semi-arid region, and then compare the results of this study, in order to highlight the characteristics of this paper.

Additional comments

It is suggested that the theme of the article highlight the characteristics of the semi-arid area.

---

## Round 0.2 · Minor Revisions

Dear authors,

your manuscript is now in a good shape. Howeve, as pointed out by the reviewer, the English needs to be improved. I checked the manuscript myself and found that this is particularly true for those parts that have undergone some changes (such as the abstract). Please initiate adequate English editing.

Kind regards
Anja Linstädter

Reviewer 1 ·

Basic reporting

no comment

Experimental design

no comment

Validity of the findings

no comment

Additional comments

Your paper is in good shape- it will provide valuable information to the scientific communities. English editing will make your paper more strong and easy to follow by readers.

---

## Round 0.3 · accepted · Accept

Dear authors,

Thanks a lot for taking care of the remaining issues, and congratulations on a fine paper. It is now ready for publication.

Kind regards
Anja Linstädter

#